# Rate of Malignant Transformation Differs Based on Diagnostic Criteria for Oral Lichenoid Conditions: A Systematic Review and Meta-Analysis of 24,277 Patients

**DOI:** 10.3390/cancers15092537

**Published:** 2023-04-28

**Authors:** Jing-Wen Li, Kar Yan Li, Bik Wan Amy Chan, Colman Patrick McGrath, Li-Wu Zheng

**Affiliations:** 1Division of Oral & Maxillofacial Surgery, Faculty of Dentistry, The University of Hong Kong, Hong Kong SAR, China; jingwen7883@gmail.com; 2Clinical Research Centre, Faculty of Dentistry, The University of Hong Kong, Hong Kong SAR, China; skyli@hku.hk; 3Department of Anatomical and Cellular Pathology, Prince of Wales Hospital, The Chinese University of Hong Kong, Hong Kong SAR, China; abwchan@cuhk.edu.hk; 4Division of Applied Oral Sciences & Community Dental Care, Faculty of Dentistry, The University of Hong Kong, Hong Kong SAR, China; mcgrathc@hkucc.hku.hk

**Keywords:** oral lichen planus, malignant transformation, oral squamous cell carcinoma, meta-analysis

## Abstract

**Simple Summary:**

Oral lichenoid conditions are common potentially malignant disorders affecting oral mucosa. The 2003 modified WHO criteria is the most widely used criteria in diagnosing oral lichen planus; however, the concern on the implementation of epithelial dysplasia as an exclusion criterion may result in the remarkable underestimation of the potential malignancy of this disease continuing. This systemic review and meta-analysis attempted to provide an objective view by comparing the malignant transformation rate of oral lichen planus diagnosed according to the 2003 modified WHO criteria and non-2003 criteria, and to determine risk variables associated with malignant transformation of oral lichenoid conditions. We found that the malignant transformation rate of oral lichen planus differed based on diagnostic criteria, with lower rates for the 2003 modified WHO criteria than non-2003 criteria, albeit not statistically significant. A higher incidence of malignant transformation was found for red-type lesions and patients who are smokers, alcohol consumers, and HCV positive.

**Abstract:**

Objectives: This systematic review and meta-analysis aims to evaluate the evidence on the malignant potential of oral lichenoid conditions (OLCs) including oral lichen planus (OLP), oral lichenoid lesions (OLL), and lichenoid mucositis dysplasia (LMD). In addition, it aims to compare the rate of malignant transformation (MT) in OLP patients diagnosed according to different diagnostic criteria, and to investigate the possible risk factors for OLP MT into OSCC. Materials and methods: A standardized search strategy was applied across four databases (PubMed, Embase, Web of Science, and Scopus). Screening, identification and reporting followed the PRISMA framework. Data on MT were calculated as a pooled proportion (PP), subgroup analyses and possible risk factors for MT were pooled as odds ratios (ORs). Results: Among 54 studies with 24,277 patients, the PP for OLCs MT was 1.07% (95% CI [0.82, 1.32]). The estimated MT rate for OLP, OLL and LMD was 0.94%, 1.95% and 6.31%, respectively. The PP OLP MT rate using the 2003 modified WHO criteria group was lower than that using the non-2003 criteria (0.86%; 95% CI [0.51, 1.22] versus 1.01%; 95% CI [0.67, 1.35]). A higher odds ratio of MT was observed for red OLP lesions (OR = 3.52; 95% CI [2.20, 5.64]), smokers (OR = 1.79; 95% CI [1.02, 3.03]), alcohol consumers (OR = 3.27, 95% CI [1.11, 9.64]) and those infected with HCV (OR = 2.55, 95% CI [1.58, 4.13]), compared to those without these risk factors. Conclusions: OLP and OLL carry a low risk of developing OSCC. MT rates differed based on diagnostic criteria. A higher odds ratio of MT was observed among red OLP lesions, smokers, alcohol consumers, and HCV-positive patients. These findings have implications for practice and policies.

## 1. Introduction

Oral lichen planus (OLP) is a T-cell-mediated chronic inflammatory disorder affecting the oral mucosa [1]. Based on its clinical presentation, OLP can be classified into non erosive-atrophic forms (reticular, popular, and plaque-like) and erosive-atrophic forms (erythematous, erosive, and bullous) [2]. The World Health Organization (WHO) proposed the diagnostic criteria of OLP in 1978, in which the oral lichenoid lesions (OLL) and lichenoid mucositis with dysplasia (LMD) were not distinguished from OLP because of the high similarity in clinical and histological characteristics [3]. Based on the findings that in response to dysplasia, mucosal lesions may develop lichen-like histological events, essentially an inflammatory lichenoid infiltrate, Krutchkoff and Eisenberg proposed an entity “oral lichenoid dysplasia (OLD)” and recommended that all cases presenting with epithelial dysplasia should be excluded from diagnosis as OLP [4]. LMD or Oral epithelial dysplasia (OED) is used in other reports to describe this entity [5,6]. In 2003, van der Meij and van der Waal proposed a modified criteria to differentiate OLP and OLL, while the presence of dysplasia is considered as a criterion to exclude OLP [7]. The modified WHO criteria are so far the most widely used criteria for diagnosing OLP. A major distinction between OLL and OLP is the underlying etiology. Whereas OLL can be triggered by (1) allergic response to dental materials, (2) usage of certain medications, (3) graft-vs-host disease (GVHD), (4) systematic disease, and (5) lesions that have a lichen-planus-like aspect, but lack one or more characteristic clinical aspects [8,9], the cause of OLP is not clear. The controversial debate on whether OLP and OLL are actually different diseases or a continuum of the same condition continues.

Moreover, it is still controversial if LMD is a distinct histopathologic entity. Farah et al. recommended that LMD should not be classed as a distinct pathological entity based on their finding that the transcriptomic and immunophenotypic profiles of LMD were similar to that of OLP but distinct from oral epithelial dysplasia [10]. In 2020, the Working Group for the WHO Collaborating Centre for Oral Cancer recommended that health professionals should not use the term “oral lichenoid dysplasia” to describe any case of OLP or lichenoid lesions that presents with dysplastic features. If dysplasia is present, the diagnosis should be OLP with dysplasia or oral epithelial dysplasia with lichenoid features [11]. Under this circumstance, the term “oral lichenoid conditions” (OLCs) is used to group together the chronic immunological disorders with controversy in diagnosis and malignant potential including OLP, OLL and LMD.

A number of studies have investigated the malignant potential of OLCs since the first case report of oral squamous cell carcinoma (OSCC) developed from OLP 1910 [12]. The reported malignant transformation (MT) of OLP is reported to vary from 0.4% to 12.5% over an average time of 5.5 years [13,14]. The MT of OLP per year ranges from 0.04% to 1.74% [15,16,17,18]. Compared to OLP, the malignant potential of OLL has been studied less extensively, varying from 1.2% to 4.4% [19]. In addition, Munoz et al. demonstrated that patients with OSCC developed from preexisting OLP lesions had a higher rate of tumor recurrence than those with primary OSCC [20]. Previous systematic reviews indicated the overall OLP MT rate ranged from 0.44% to 1.37% [21,22,23]. However, the misdiagnosis of the OLP based on the multistandard diagnostic criteria was rarely taken into consideration. Furthermore, OLCs have not been studied to investigate the malignant potential holistically. The geographic variation, inconsistent diagnostic criteria, and lack of reliable prediction model have contributed to the controversy in the risk of MT. Moreover, as a chronic, complex, multi-step progression, no consensus exists regarding the risk factors from OLCs to OSCC because the majority of studies have hitherto provided limited information on variables that might influence malignization.

The objective of this systematic review and meta-analysis was to investigate the malignant potential of OLCs, to compare the rate of MT of OLP based on the diagnostic criteria of the 2003 modified WHO criteria and non-2003 modified WHO criteria, and to determine risk variables associated with MT of OLCs.

## 2. Materials and Methods

The systematic review and meta-analysis were performed according to the Preferred Reporting Items for Systematic Reviews and Meta-Analysis (PRISMA) and Meta-analysis of Observational Studies in Epidemiology (MOOSE) guidelines.

### 2.1. Search Strategy and Study Selection

PubMed, Embase, Web of Science and Scopus were searched to identify eligible studies. Title, abstract and keywords of studies were searched using free terms and combined database thesaurus terms including Medical Subject Headings (MeSH) and Embase Subject Headings (Emtree) to maximize sensitivity (Table 1). Two reviewers independently reviewed the titles and abstracts of eligible studies, and the selected papers were re-evaluated in full text.

### 2.2. Eligibility Criteria

Inclusion criteria: (1) Full-length, peer-reviewed original research papers published in English from January 2003 to September 2022; (2) Observational studies providing sufficient information on the diagnosis and rate of MT of OLCs.

Exclusion criteria: (1) Letters to the editor, case reports, reviews, book chapters and any study in a language other than English; (2) Studies that do not differentiate between OLP and cutaneous LP or lesions from other anatomical sites; (3) In vitro or animal studies, and cross-sectional clinical studies without providing the data of follow-up and MT.

### 2.3. Data Extraction and Management

The following data from each included study were extracted: (1) year and region of the study, study design, type of lesions and follow-up details including range, mean, median and frequency; (2) risk of bias assessment domains; (3) diagnostic criteria; (4) clinical patterns of OLP including (red: erosive/atrophic/bullous; white: reticular/plaque/papular); (5) characteristics of MT occurrence including: c. location; b. developed from which type of OLCs (OLP, OLL or LMD) or sub-type of OLP (white or red lesions); (6) associated risk factors of MT; (7) detailed descriptions of confirmed OLP/OLL/LMD-related malignancies. Disagreements between reviewers were resolved by consensus or by the decision of a third independent reviewer.

### 2.4. Risk of Bias Assessment

Newcastle–Ottawa scale was used to assess the risk of bias within the included observational studies with the following modifications:

Selection: (1) A cohort of patients that is representative of the general population; (2) the diagnosis of OLCs must be based on both clinical and histopathological confirmation; (3) cases of OLP/OLL and epithelial dysplasia must be well differentiated.

Comparability of groups on demographic characteristics and important potential confounders: While presenting variables from the study cohort, risk factors including smoking and alcohol consumption must be documented.

Outcomes: (1) Independent or blind assessment stated in the paper; (2) all variables associated with MT must be documented in details including age, gender, clinical sub-type, location of lesions and tumors, follow-up information, TNM, related risk factors and intervention; (3) sufficient follow-up duration (2 years or more); (4) adequacy of follow-up of cohorts.

In view of the design of the studies included without non-exposed group, we modified the Newcastle–Ottawa Scale to suit the selected studies by eliminating question No.2 (the selection of the non-exposed group). In this case, the total Newcastle–Ottawa Scale has been revised to be out of eight, rather than nine [21]. Each study can be awarded a maximum of one check mark for each numbered item. The maximum number of check mark is 8. Two reviewers performed parallel independent assessments of each article. Disagreements were resolved by consensus among both reviewers; if no agreement could be reached, a third independent reviewer would decide.

### 2.5. Evaluation of Quality of Evidence

The quality of evidence was evaluated using the Grading of Recommendations Assessment, Development and Evaluation (GRADE) system. The quality of evidence was classified in one of four levels: very low, low, moderate or high. The overall quality rating was based on the following domains: risk of bias, inconsistency, indirectness, imprecision and publication bias [24].

### 2.6. Data Analysis

The statistical analysis was performed using the software Stata (StataCorp. 2019. Stata Statistical Software: Release 16. StataCorp LLC., College Station, TX, USA) with user-written commands. *p* < 0.05 was considered significant. Cumulative meta-analysis was performed calculating the pooled proportion of the overall risk of MT in OLCs and the subtype of OLP, OLL and LMD under the random effects model using Der-Simonian Liard method. Heterogeneity was evaluated using Cochran’s Q statistic and quantified using Higgin’s I^2^ statistics. Subgroup analysis was conducted to compare the difference in MT based on the diagnostic criteria, regional disparity and location of OLP. To investigate the possible risk factors that could potentially lead to the malignization of OLCs, we performed a comprehensive analysis based on gender of patients, clinical sub-type, habits of smoking or alcohol consumption, infection with hepatitis C virus (HCV), diabetes mellitus and hypertension. The odds ratio (OR), along with 95% confidence interval (CI), was applied for the representation of pooled results from studies. Publication bias was assessed using a funnel plot and quantified using Egger’s linear regression test. Asymmetry of the collected studies’ distribution by visual inspection or *p*-value < 0.1 indicated publication bias [25].

## 3. Results

### 3.1. Result of Database Search

A total of 5596 articles published between 2003 and 2022 were retrieved. After eliminating the duplicate records, 3039 papers were selected for title and abstract evaluation. Sixty-seven publications were checked by full-text review, of which 13 studies did not meet all inclusion criteria. A total of 54 studies were deemed eligible for inclusion in the final meta-analysis. Figure 1 shows the flowchart of the study selection process recommended by the PRISMA Statement. Baseline data and characteristics of the included studies are presented in Appendix A.

### 3.2. Study Characteristics

The included studies comprised 54 cohorts; 48 of them harbored a retrospective design [19,26,27,28,29,30,31,32,33,34,35,36,37,38,39,40,41,42,43,44,45,46,47,48,49,50,51,52,53,54,55,56,57,58,59,60,61,62,63,64,65,66,67,68,69,70,71,72], and 6 were prospective [5,17,73,74,75,76]. A total of 24,277 patients were included: 22,578 with OLP, 717 with OLL, 153 with LMD, and 829 patients in one study without differentiating OLP and OLL [72]. Two-thirds of the included patients were female (Male: Female ratio was 1:1.893). Age was heterogeneously reported in the studies and ranged from 6 to 98 years old. The information regarding mean age, available in 45 studies analyzing 18,632 patients, had a value of 54.34 years. The mean follow-up period ranged from 16.8 to 272.4 months. There were 29 studies from Europe, 17 from Asia, 4 from America, 2 from Africa, and 2 from Oceania.

### 3.3. Rate of Malignant Transformation

Among a total of 24,277 patients, 331 cases progressed to OSCC. The timing of MT progression, calculated as the time from first diagnosis of OLP/OLL/LMD until the onset of the carcinoma, was demonstrated in 36 out of 54 studies, based on the average follow-up periods ranging from 3 to 108 months. The pooled proportion meta-analysis showed that the overall MT rate across all the OLCs cases was 1.07 [95% CI = 0.82–1.32]. There was a considerable heterogeneity among the results of individual studies, as indicated by an I^2^ value of 71.10% and Chi-Square *p* < 0.001 (Figure 2). The annual and monthly transformation rates was 0.32% and 0.027%, respectively. Synthesized data on MT rate of included studies are presented in Table 2.

In OLP group (Figure 3a), the MT rate was 0.94% [95% CI = 0.70–1.19]. The heterogeneity was significant (*p* < 0.001) along with a considerable degree (I^2^ = 67.3%). OLL group (Figure 3b) showed an MT rate of 1.95% [95% CI = 0.15–3.75]. The heterogeneity was not significant (*p* = 0.064), but a considerable degree was found (I^2^ = 52.1%). In LMD group (Figure 3c), the MT rate was 6.31% [95% CI = 2.30–10.32] with no heterogeneity observed (I^2^ = 0%). Pairwise meta-regression analysis revealed significant difference in MT between the OLP and LMD groups (*p* = 0.023). From the regional variation point of view, the highest MT was found in Africa (4.04%), followed by North America (2.71%), Oceania (2.24%), Europe (1.29%) and Asia (0.61%), while South America reported no data on MT (Figure 4).

In addition, a subgroup analysis comparing the MT rate in patients diagnosed using the 2003 modified WHO criteria (22 studies) with that in patients using the non-2003 modified WHO criteria (29 studies) was performed. The pooled proportion showed the OLP MT rate of the 2003 modified WHO criteria group was lower than the non-2003 group (0.86% (95% CI [0.51, 1.22]) vs. 1.01% (95% CI [0.67, 1.35])). The substantial heterogeneity was identified in both groups with an I^2^ of 64.3% and 67.3%, respectively (Figure 5). However, the pairwise meta-regression analysis demonstrated that there was no significant difference between the sub-groups (*p* = 0.667). Furthermore, with respect to the site of OLP, the most common location of transformation in our meta-analysis was on buccal mucosa (0.77%; 95% CI [0.27, 1.27]), followed by tongue (0.67%; 95% CI [0.10, 1.25]) (Figure 6a,b).

### 3.4. Meta-Analysis for Risk Factor of Malignant Transformation

Thirty-one studies with 19,388 OLP patients reported the difference in MT rate between genders. Females had a higher rate of MT (OR = 1.18; 95% CI [0.90, 1.55]); however, the difference was not significant (*p* = 0.241; Figure 7). Red lesions (erosive, atrophic, bullous) demonstrated a significantly higher MT rate compared to the white lesions (reticular, plaque, papular) (OR = 3.52; 95% CI [2.20, 5.64]; *p* < 0.001; Figure 8a). Fourteen studies with 7594 patients provided data on MT rate in smokers and non-smokers. MT rate was significantly higher in smoking patients (OR = 1.79; 95% CI [1.02, 3.03]; *p* = 0.043; Figure 8b). Similar result was found on the influence of habit of alcohol consumption on MT. Pooled data from 7 studies with 2775 patients showed the MT rate in alcohol consumption group was significantly higher than that in non-alcohol consumption group (OR = 3.27, 95% CI [1.11, 9.64], *p* = 0.031; Figure 8c). Data from 10 homogenous studies (I^2^ = 0%) with 6850 patients revealed a significantly higher risk of MT in HCV-positive patients in comparison to the HCV-negative patients (OR = 2.55, 95% CI [1.58, 4.13], *p* < 0.001; Figure 8d). Patients with diabetes mellitus (DM) or hypertension showed no statistical difference with patients without diabetes (OR = 1.89, 95% CI [0.80, 4.44], *p* = 0.146; Figure 8e) or hypertension (OR = 1.48, 95% CI [0.22, 10.25], *p* = 0.689; Figure 8f).

### 3.5. Risk of Bias Assessment and Quality of Evidence

Among the 54 studies, the NOS scale score ranged between 3 and 8. Overall, 28 studies scored ≥6 and 16 studies scored 5, while only one study scored 3. Only seven out of fifty-four studies fulfill the adequate follow-up duration (2 years or more). Full details of the risk of bias assessment are shown in Appendix A. The visual inspection analysis of the asymmetry of the funnel plot and the statistical test performed for the same purpose (pEgger < 0.001) revealed that the publication bias cannot be completely rule out among other sources of bias (Appendix A). Based on GRADE system, our systematic review allowed us to obtain OLP, OLL and LMD MT estimations with a moderate-to-high quality of evidence (Appendix A).

## 4. Discussion

This systematic review revealed an overall MT rate of 1.07% in OLCs based on the analysis of 54 studies with a total of 24,277 patients. Overall, 0.94% of OLP patients developed OSCC, which represents the lowest risk among the OLCs, while the MT rate of OLL and LMD was 1.95% and 6.31%, respectively.

The presence of epithelial dysplasia is considered as the gold-standard indicator for the assessment of cancer risk and progression in OPMD. Whether epithelial dysplasia demonstrates OLP/OLL-like features, or OLP/OLL lesions gain dysplastic features during the progress of the disease, is a continuing debate. Some clinical and pathological evidence, including our recent study, revealed that OLP may develop epithelial dysplasia along the course of its evolution. In a recent case study, we found that one patient with OLP (no dysplasia at the time of initial biopsy) developed well-differentiated OSCC and moderate epithelial dysplasia at two different sites of long-standing lichen planus at the tongue. This finding supports the concept that OLP may develop dysplastic features during the progress of the disease, which may finally progress to OSCC. While solid evidence is still lacking, many clinicians and pathologists raised the issue of whether or not the detection of epithelial dysplasia invalidates the diagnosis of OLP. The poorly understood pathogenesis of OLP leads to the inconsistency in diagnostic criteria. While the 2003 modified WHO criteria have become the current consensus for the diagnosis of OLP, the consideration of epithelial dysplasia as an exclusion criterion may result in a remarkable underestimation of the potential malignancy of this disease [77]. González-Moles et al. reported in a meta-analysis that although non-dysplastic OLP may progress to cancer in 0.44–2.28% of the cases, the malignancy in OLP with dysplasia is significantly higher (6.22%) [78]. In a selective pooled proportion analysis of studies using the 2003 modified WHO diagnostic criteria (22 studies with a total of 10,728 patients) in our study, the MT rate (0.86%) was lower than the studies (29 studies with a total of 11,850 patients) using the non-2003 criteria (1.01%), but the difference was not significant (*p* = 0.667). To our knowledge, this is the first study comparing the MT rate of OLP patients diagnosed with different diagnostic criteria. It is worth noting that our meta-analysis revealed that the MT rate of LMD was significantly higher than that of OLP (*p* = 0.023) Our findings advocate the suggestion from González-Moles et al.: “Further studies and a revision of the diagnostic criteria of the disease are required to establish the true prevalence of this important risk marker for cancer progression” [79].

While OLP and OLL tend to be considered as distinct conditions in the diagnostic criteria proposed by van der Meij and van der Waal, they do share significant overlapping features [7,80]. In a prospective study, Van der Meij et al. [73] revealed an MT rate of 1.7% in OLL patients, whereas MT was not found in any OLP patients. González-Moles et al. [78] demonstrated no significant differences in MT between OLP (2.28%) and OLL (2.11%), and the authors suggested that they should not be considered as different entities. Our study revealed that the MT risk of OLL was higher than that of OLP, albeit with no statistical significance (*p* = 0.728).

To investigate the risk factors that could potentially prompt the malignization of OLP, we conducted a comprehensive analysis on the variables including the gender, clinical patterns, smoking and alcohol consumption, HCV, diabetics and hypertension based on the available data. Aghbari et al. reported that erosive-type OLP carries the highest risk of MT (1.7%), followed by atrophic (1.3%) and reticular (0.1%) patterns [81]. Laniosz et al. revealed that erosive-type OLP was the most prevalent clinical sub-type for MT, accounting for 71.4% of cases that developed OSCC from OLP [66]. Tsushima et al. highlighted that the mean duration of MT was much shorter in red-type OLP than in white-type OLP [69]. In our meta-analysis, red-type OLP (atrophic, erosive, bullous) carries a significantly higher risk of developing OSCC than white lesions (reticular, papular, plaque) (*p* < 0.001). The upregulation of the expression of p53 and matrix metalloproteinases, the potential cancer biomarkers in red-type OLP, provides basic evidence of mechanism of the cancer development [82,83].

Consistently with some other studies [81,84], our meta-analysis showed that the prevalence of MT was significantly higher in tobacco (OR_smoke_ = 1.79, P_smoke_ = 0.043) and alcohol users (OR_alcohol_ = 3.27, P_alcohol_ = 0.031) compared to those patients without the habits. Smoking is categorized as a major risk factor of oral cancer. It is highly debatable whether smoking independently causes OSCC or just interacts with OLP to increase its malignant potential in the OLP-OSCC patient. Therefore, as part of the 2003 WHO modified criteria, smoking documentation must be in placed in order to identify the true MT. Alcohol may promote malignancy through atrophying the oral epithelium and increasing the permeability of epithelial cells. Chronic use of alcohol has been reported to have genotoxic and mutagenic effects and to impair both innate and acquired immunity, increasing the risk of infections and cancers [85]. Meanwhile, some studies emphasize the need for the patients with habits of smoking and alcohol consumption to avoid or discontinue these habits in order to reduce the risk of MT [86,87].

Another risk factor for OLP malignization to emerge in this meta-analysis was the presence of HCV infection. HCV-positive patients showed significantly higher MT prevalence compared to the HCV-negative patients (*p* < 0.001). Rangel et al. reported that 12.8% of head and neck cancer patients were HCV positive [88]. Several studies have demonstrated that HCV infection is strongly associated with OLP malignization, such that it is interpreted that the virus may replicate in the oral mucosa and attract HCV-specific T lymphocytes [89,90]. Nagao et al. revealed that OLP occurrence in HCV-infected individuals is not only associated with the virus itself, but also with other host factors including immunity, genetics and insulin resistance [91]. Moreover, Gandolfo et al. found that 44% of OLP patients who development carcinomas were HCV positive and suggested that HCV infection apparently increased the risk in the neoplastic transformation of OLP [29]. However, the debate continues as to whether HCV induces carcinogenesis directly through the cell cycle deregulation or indirectly by causing liver cirrhosis [92,93]. This notwithstanding, we believe that the HCV screening is essential in patients with OLP. Treatment interventions for HCV infection may reduce the likelihood of developing hepatocellular carcinoma and OSCC.

Systemic diseases such as DM and hypertension are known to increase the risk of significant oral problems, which are often detected in cases of OLP [39,75,94]. The pathophysiology of hypertension and cancer is intertwined, and multiple studies have established an association between DM and oral cancer [95,96,97,98]. Ramos-Garcia et al. demonstrated that patients with DM have higher odds of developing oral cancer and OPMD than non-diabetic patients. Moreover, oral cancer patients with DM have a higher mortality rate compared to those without DM [99]. Although the current meta-analysis did not identify significant differences, we recommend that clinicians improve the management of OLP patients with DM and hypertension. Establishing follow-up protocols for specific populations can help reduce the risk of oral cancer progression and associated healthcare costs.

### 4.1. Strengths

To the best of our knowledge, this is the most comprehensive systematic review and meta-analysis to focus on the MT rate and potential risk factors of OLCs. We attempt to overcome the potential diagnostic confusion on OLP, OLL and LMD by categorizing the cases into different analytic datasets to calculate the MT of each pathological condition independently.

It has been 20 years since the 2003 modified WHO criteria was Introduced, and they have become the most widely used criteria in diagnosing OLP. While concerns remain that the implementation of epithelial dysplasia as an exclusion criterion may result in a remarkable underestimation of the potential malignancy of this disease, we attempt to provide an objective view by comparing the MT rate of OLP patients diagnosed according to the 2003 modified WHO criteria and non-2003 criteria. The result demonstrates that the MT rate calculated from the OLP patients diagnosed with the criteria proposed by van der Meij and van der Waal is lower than that with the non-2003 criteria, but the difference was not significant.

### 4.2. Limitations

There are several potential limitations associated with our study. Firstly, we only included studies published in the English language. Secondly, despite our efforts, it may not be fully possible to distinguish OLP, OLL and LMD in the included studies. Third, not all studies provided numerical data of associated risk factors, which may limit the implementation of our meta-analysis in identifying the role of MT in OLP. Finally, the follow-up periods among the included studies were highly variable, leaving the possibility that some OLP/OLL patients would have progressed to OSCC if followed up for a longer period.

### 4.3. Recommendations for Future Research and Clinical Practice

Ideally, to investigate the MT risk of oral potentially malignant disorders is to conduct a prospective clinical trial with a regular and long follow-up. However, practically speaking, this is very difficult due to the highly variable risk and lengthy time interval of the transition of premalignancy to OSCC. Considering the difficulties in conducting a prospective clinical trial, a large-scale multicenter cohort study would be the optimal option to address this issue. Proper clinical and histopathological documentation, including careful differentiation between OLP and other lichenoid lesions, and detailed numerical data of associated risk factors, should be emphasized in future research. As the OLP is considered to be premalignant, an evidence-based recall system would be useful to improve the chance of early diagnosis of oral cancer; however, the evidence regarding the frequency of follow-up/recall is very small and far from reliable due to controversy on the risk of MT. The majority of the reported OPMD risk assessment models are based on conventional statistical approaches. Machine learning algorithms represent an advancement to conventional statistical models for evaluating the risk of MT. The deep learning algorithm may not only provide an effective and reliable prediction and decision-making model for clinicians to accurately estimate the risk of MT, but it also builds a solid ground for future development and implementation of a user-friendly web-based platform not only for specialists but also for dentists and family doctors planning management and monitoring protocols. In addition, efficacious communication with patients is also vitally important to raise public awareness about the malignant potential of the oral lichenoid condition.

## 5. Conclusions

The systemic review and meta-analysis provide evidence that OLP and OLL carries a low risk of developing OSCC. The MT rate of OLP differed based on diagnostic criteria, with lower MT rates for the 2003 modified WHO criteria than non-2003 criteria, albeit not statistically significant. A higher incidence of MT was found for red OLP lesions and patients who are smokers, alcohol consumers, and HCV positive. The mechanism dictating how these risk factors promote MT needs further clarification. An evidence-based recall system would be useful to improve the chance of early diagnosis of oral cancer, but such a system is not yet available. The deep learning algorithm and web-based platform may provide a reliable and user-friendly prediction and decision-making model not for only specialists but also for dentists and family doctors planning management and monitoring protocols.

## Figures and Tables

**Figure 1 cancers-15-02537-f001:**
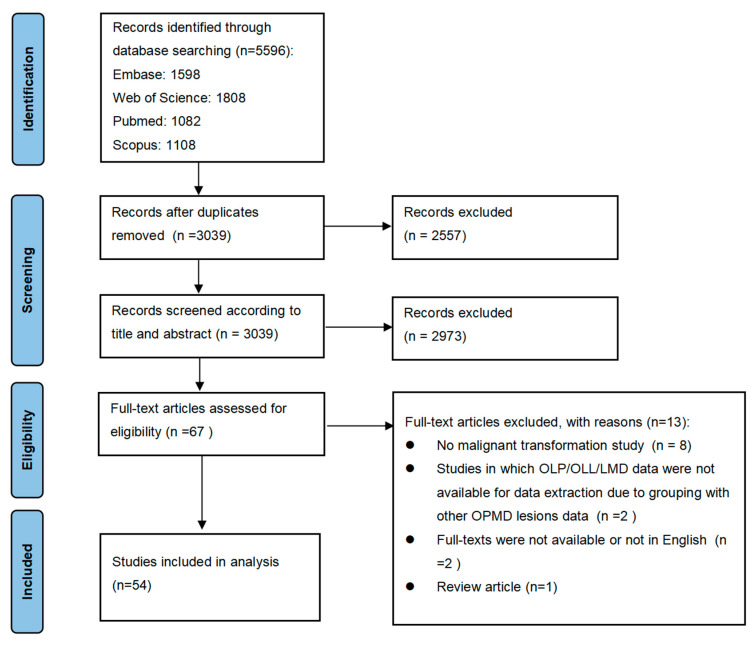
A flow diagram of the literature search and study selection process.

**Figure 2 cancers-15-02537-f002:**
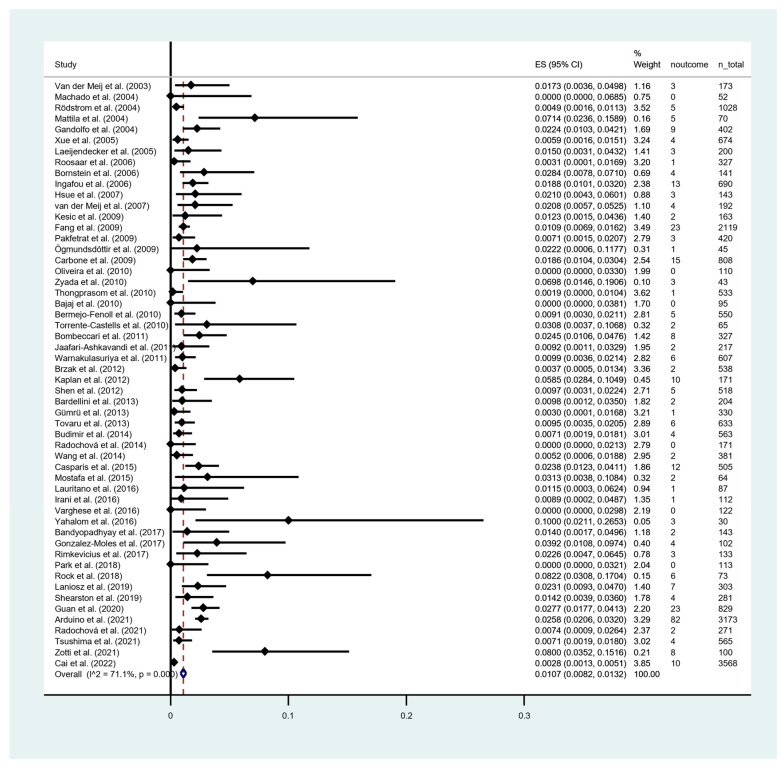
Forest plot of meta-analysis on overall malignant transformation rate of oral lichenoid conditions. I^2^, the statistics for heterogeneity; ES, estimation; CI, confidence interval; noutcome, event transformation into OSCC; *n*_total, total number of patients [5,17,19,26,27,28,29,30,31,32,33,34,35,36,37,38,39,40,41,42,43,44,45,46,47,48,49,50,51,52,53,54,55,56,57,58,59,60,61,62,63,64,65,66,67,68,69,70,71,72,73,74,75,76].

**Figure 3 cancers-15-02537-f003:**
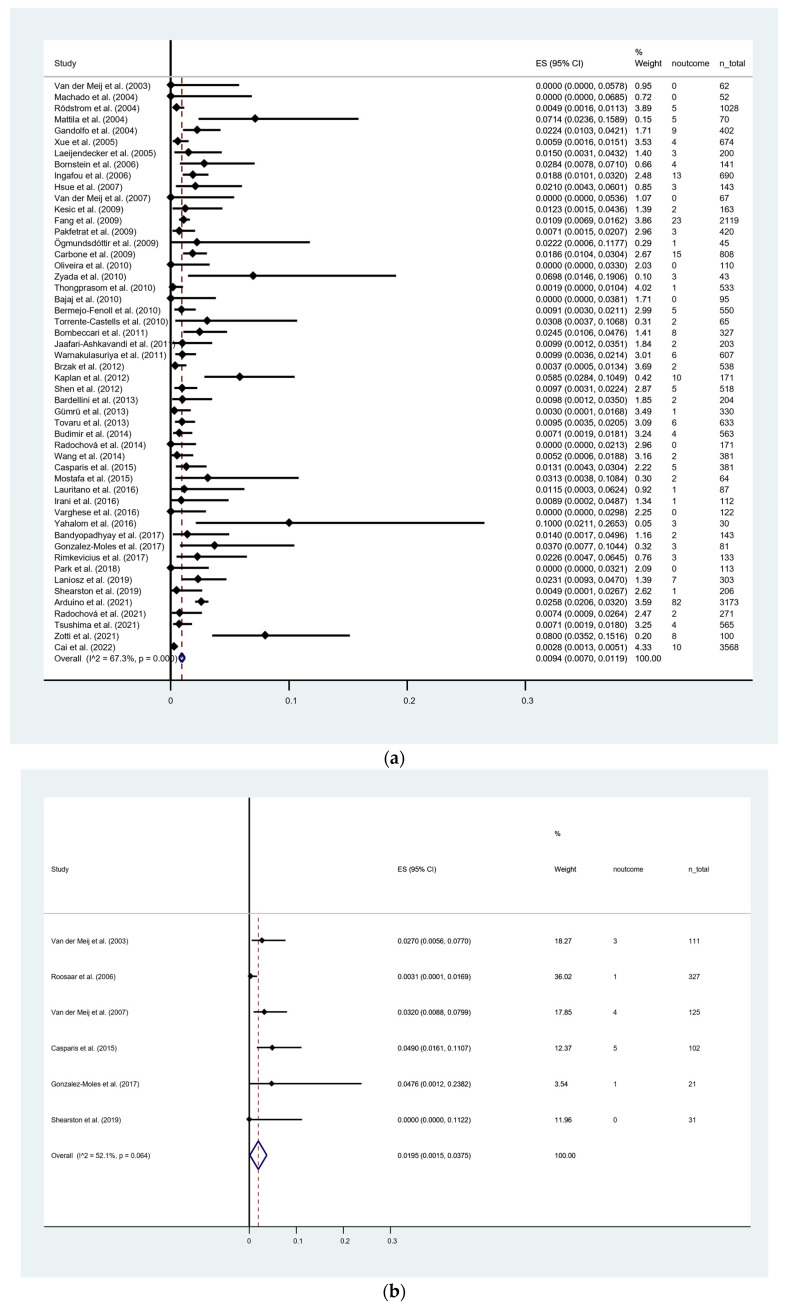
(**a**) Forest plot of sub-group meta-analysis on malignant transformation rate of oral lichen planus. (**b**) Forest plot of sub-group meta-analysis on malignant transformation rate of oral lichenoid lesions. (**c**) Forest plot of sub-group meta-analysis on malignant transformation rate of lichenoid mucositis with dysplasia. I^2^, the statistics for heterogeneity; ES, estimation; CI, confidence interval; noutcome, event transformation into OSCC; *n*_total, total number of patients [5,17,19,26,27,28,29,30,31,32,33,34,35,36,37,38,39,40,41,42,43,44,45,46,47,48,49,50,51,52,53,54,55,56,57,58,59,60,61,62,63,64,65,66,67,68,69,70,71,72,73,74,75,76].

**Figure 4 cancers-15-02537-f004:**
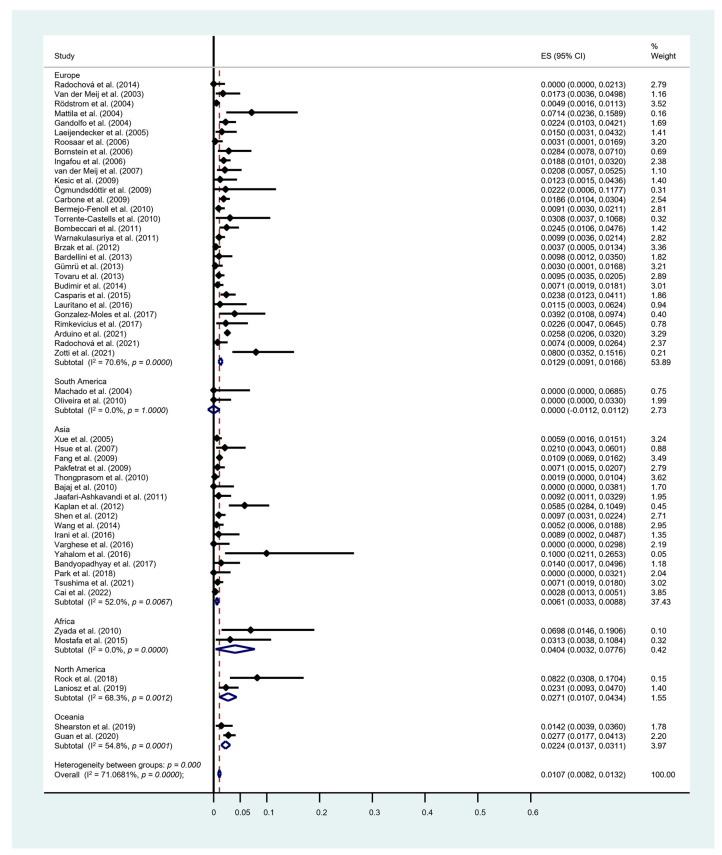
Forest plot of meta-analysis on malignant transformation rate stratified by regional distribution. I^2^, the statistics for heterogeneity; ES, estimation; CI, confidence interval [5,17,19,26,27,28,29,30,31,32,33,34,35,36,37,38,39,40,41,42,43,44,45,46,47,48,49,50,51,52,53,54,55,56,57,58,59,60,61,62,63,64,65,66,67,68,69,70,71,72,73,74,75,76].

**Figure 5 cancers-15-02537-f005:**
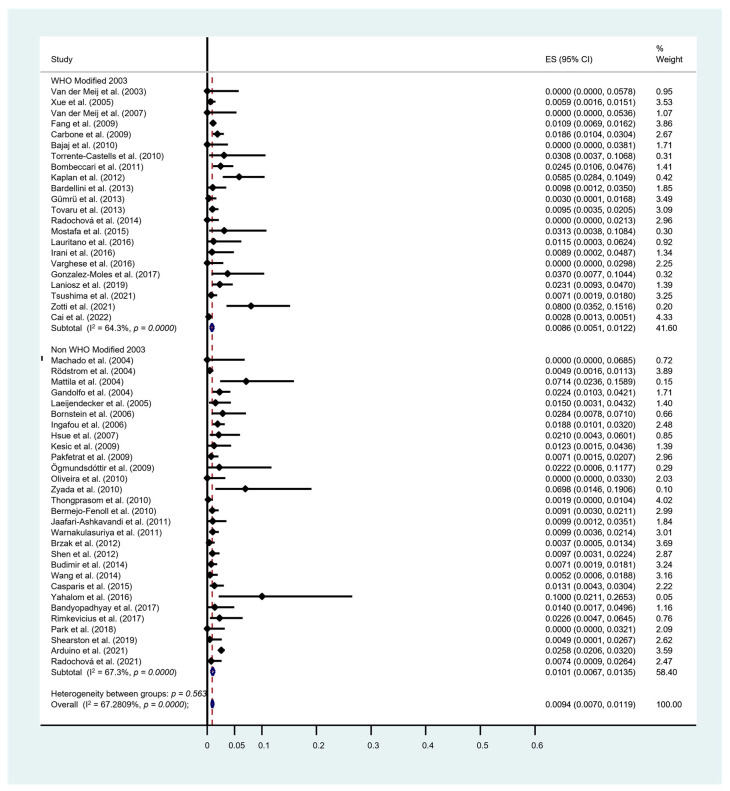
Forest plot of sub-group meta-analysis on malignant transformation rate based on the 2003 and non-2003 WHO modified diagnostic criteria, respectively. I^2^, the statistics for heterogeneity; ES, estimation; CI, confidence interval; noutcome, event transformation into OSCC [17,19,26,27,28,29,30,31,32,33,34,35,36,37,38,39,40,41,42,43,44,45,46,47,48,49,50,51,52,53,54,55,56,57,58,59,60,61,62,63,64,65,66,67,68,69,70,71,73,75,76].

**Figure 6 cancers-15-02537-f006:**
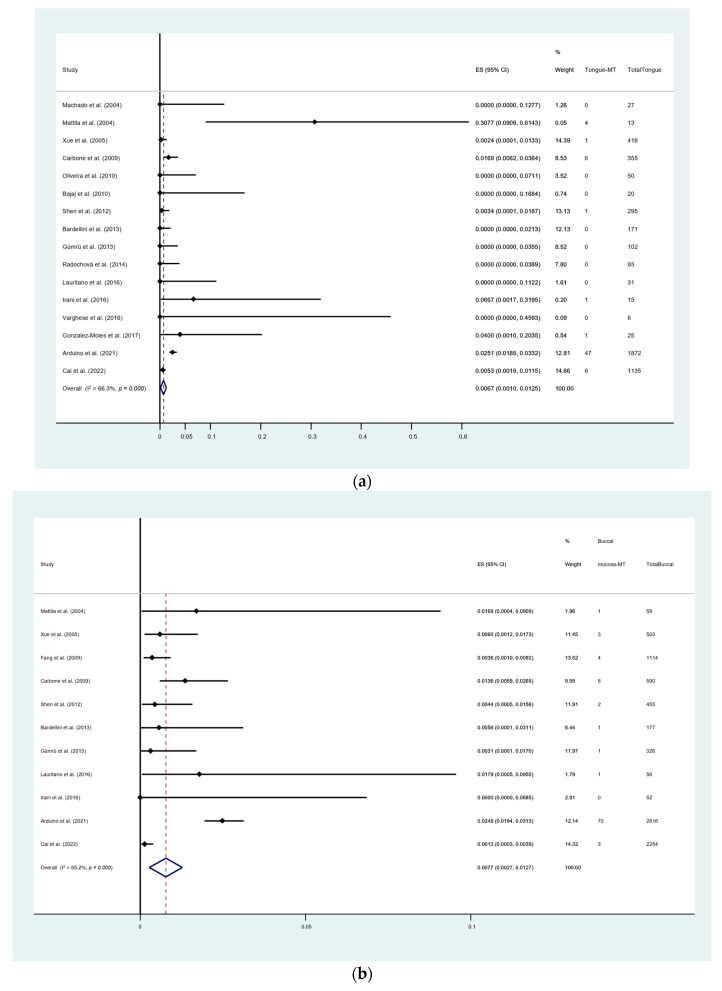
(**a**) Forest plot of sub-group meta-analysis on malignant transformation rate of oral lichen planus with the site of tongue. (**b**) Forest plot of sub-group meta-analysis on malignant transformation rate of oral lichen planus with the site of buccal mucosa. I^2^, the statistics for heterogeneity; ES, estimation; CI, confidence interval [26,28,30,36,39,40,43,50,51,52,55,59,60,61,63,67,71].

**Figure 7 cancers-15-02537-f007:**
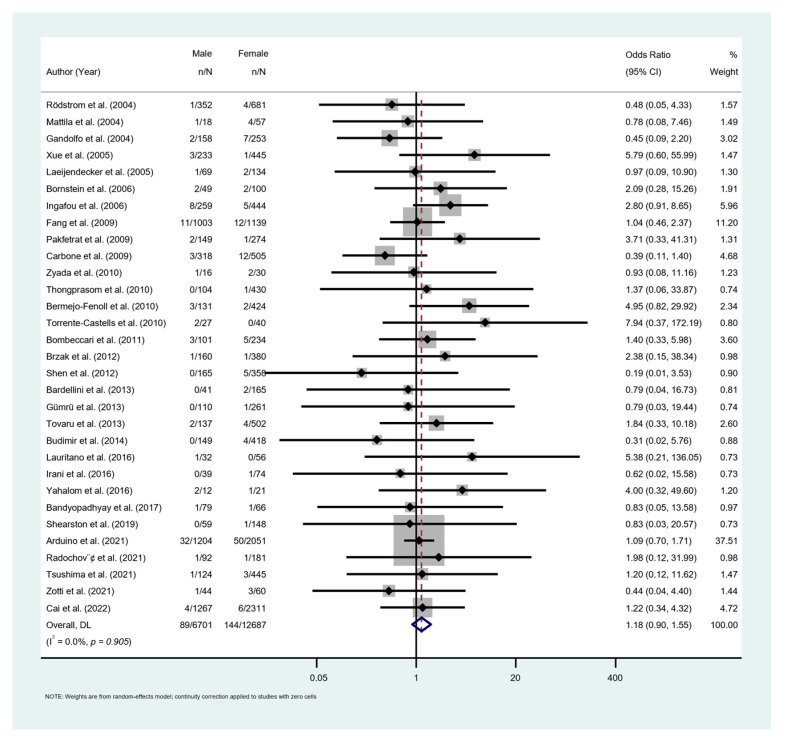
Secondary meta-analysis. Malignant transformation rate by related risk factors. Forest plots for odds ratios of the association between OLP malignant transformation and gender [19,27,28,29,30,31,32,33,34,36,37,39,41,42,44,45,48,50,51,52,53,54,59,60,62,67,68,69,70,71,73,74,75,76].

**Figure 8 cancers-15-02537-f008:**
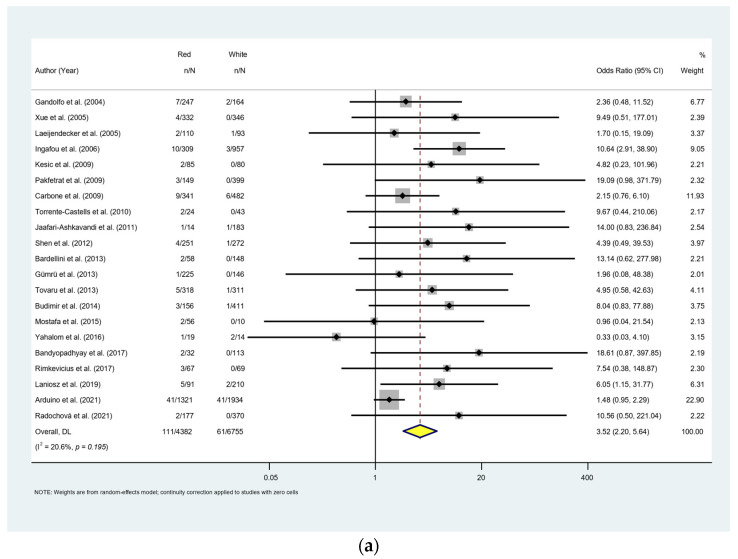
(**a**) Secondary meta-analysis. Malignant transformation rate by related risk factors. Forest plots for odds ratios of the association between OLP malignant transformation and clinical sub-types. (**b**) Secondary meta-analysis. Malignant transformation rate by related risk factors. Forest plots for odds ratios of the association between OLP malignant transformation and smoking. (**c**) Secondary meta-analysis. Malignant transformation rate by related risk factors. Forest plots for odds ratios of the association between OLP malignant transformation and alcohol consumption. (**d**) Secondary meta-analysis. Malignant transformation rate by related risk factors. Forest plots for odds ratios of the association between OLP malignant transformation and HCV infection. (**e**) Secondary meta-analysis. Malignant transformation rate by related risk factors. Forest plots for odds ratios of the association between OLP malignant transformation and diabetes mellitus. (**f**) Secondary meta-analysis. Malignant transformation rate by related risk factors. Forest plots for odds ratios of the association between OLP malignant transformation and diabetes mellitus [19,29,30,31,32,33,35,37,39,41,42,44,45,46,50,51,52,53,54,58,62,64,66,67,68,69,70,75,76].

**Table 1 cancers-15-02537-t001:** Systematic review search strategy for PubMed, Embase, Web of Science and Scopus.

Database	Search Terms
PubMed	(“Lichen Planus, Oral”[Mesh] or “oral lichen planus”[All Fields] or “oral lichenoid lesion”[All Fields] or “olp”[All Fields] or “oll”[All Fields]) and (“Carcinoma, Squamous Cell”[Mesh] or “squamous cell carcinoma”[All Fields] or “oscc”[All Fields] or “precancer”[All Fields] or “cancer”[All Fields] or “neoplasm”[Mesh] or “neoplasm”[All Fields] or “oral potentially malignant disorder”[All Fields] or “premalignant”[All Fields] or malign * or premalign * or “malignant transformation”[All Fields] or “degeneration”[All Fields] or “carcinogenesis”[Mesh] or “carcinogenesis”[All Fields] or “transformation”[All Fields] or “oncogenesis”[All Fields])
Embase	(‘oral lichen planus’/exp OR ‘oral lichen planus’ OR ‘oral lichenoid lesion’/exp OR ‘oral lichenoid lesion’ OR ‘OLP’ OR ‘OLL’) AND (‘oral squamous cell carcinoma’/exp OR ‘oral squamous cell carcinoma’ OR ‘OSCC’ OR ‘precancer’/exp OR ‘precancer’ OR ‘cancer’ OR ‘malignant neoplasm’/exp OR ‘malignant neoplasm’ OR ‘oral potentially malignant disorder’/exp OR ‘oral potentially malignant disorder’ OR ‘premalignant’ OR ‘malign *’ OR ‘premalign *’ OR ‘malignant transformation’/exp OR ‘malignant transformation’ OR ‘degeneration’/exp OR ‘degeneration’ OR ‘carcinogenesis’/exp OR ‘carcinogenesis’ OR ‘oncogenesis’)
Web of Science	(ALL = (oral lichen planus) OR ALL = (oral lichenoid lesion) OR ALL = (OLL) OR ALL = (OLP)) AND (ALL = (oral squamous cell carcinoma) OR ALL = (oscc) OR ALL = (precancer) OR ALL = (cancer) OR ALL = (malignant neoplasm) OR ALL = (oral potentially malignant disorder) OR ALL = (premalignant) OR ALL = (malign *) OR ALL = (premalign *) OR ALL = (malignant transformation) OR ALL = (degeneration) OR ALL = (carcinogenesis) OR ALL = (transformation) OR ALL = (oncogenesis))
Scopus	TITLE-ABS-KEY((“oral lichen planus” OR “oral lichenoid lesion” OR “olp” OR “oll”) AND (“oral squamous cell carcinoma” OR “oscc” OR “precancer” OR “cancer” OR “malignant neoplasm” OR “oral potentially malignant disorder” OR “premalignant” OR “malign *” OR “premalign *” OR “malignant transformation” OR “degeneration” OR “carcinogenesis” OR “transformation” OR “oncogenesis”))

**Table 2 cancers-15-02537-t002:** Synthesized data on malignant transformation rate.

Study/Year		Study Design	Total (*n*)	OSCC Development	Overall MT Rate (%)	Annual MT Rate (%)	Monthly MT Rate (%)
Van der Meij et al. [73]	2003	P	173	3	1.73	0.65	0.054
Machado et al. [26]	2004	R	52	0	0.00	0.00	0.000
Rödstrom et al. [27]	2004	R	1028	5	0.49	0.08	0.006
Mattila et al. [28]	2004	R	70	5	7.14	2.61	0.218
Gandolfo et al. [29]	2004	R	402	9	2.24	0.60	0.050
Xue et al. [30]	2005	R	674	4	0.59	0.07	0.005
Laeijendecker et al. [31]	2005	R	200	3	1.50	4.19	0.349
Roosaar et al. [74]	2006	P	327	1	0.31	0.01	0.001
Bornstein et al. [32]	2006	R	141	4	2.84	1.32	0.110
Ingafou et al. [33]	2006	R	690	13	1.88	-	-
Hsue et al. [34]	2007	R	143	3	2.10	1.72	0.143
Van der Meij et al. [17]	2007	P	192	4	2.08	0.63	0.052
Kesic et al. [35]	2009	R	163	2	1.23	-	-
Fang et al. [36]	2009	R	2119	23	1.09	0.81	0.068
Pakfetrat et al. [37]	2009	R	420	3	0.71	0.43	0.036
Ögmundsdóttir et al. [38]	2009	R	45	1	2.22	0.14	0.012
Carbone et al. [39]	2009	R	808	15	1.86	0.43	0.035
Oliveira et al. [40]	2010	R	110	0	0.00	-	-
Zyada et al. [41]	2010	R	43	3	6.98	-	-
Thongprasom et al. [42]	2010	R	533	1	0.19	0.01	0.001
Bajaj et al. [43]	2010	R	95	0	0.00	-	-
Bermejo-Fenoll et al. [44]	2010	R	550	5	0.91	-	-
Torrente-Castells et al. [45]	2010	R	65	2	3.08	-	-
Bombeccari et al. [75]	2011	P	327	8	2.45	0.75	0.062
Jaafari-Ashkavandi et al. [46]	2011	R	217	2	0.92	-	-
Warnakulasuriya et al. [47]	2011	R	607	6	0.99	-	-
Brzak et al. [48]	2012	R	538	2	0.37	-	-
Kaplan et al. [49]	2012	R	171	10	5.85	-	-
Shen et al. [50]	2012	R	518	5	0.97	0.16	0.014
Bardellini et al. [51]	2013	R	204	2	0.98	0.24	0.020
Gümrü et al. [52]	2013	R	330	1	0.30	0.15	0.013
Tovaru et al. [53]	2013	R	633	6	0.95	-	-
Budimir et al. [54]	2014	R	563	4	0.71	0.09	0.008
Radochová et al. [55]	2014	R	171	0	0.00	0.00	0.000
Wang et al. [56]	2014	R	381	2	0.52	0.78	0.065
Casparis et al. [57]	2015	R	505	12	2.38	1.50	0.125
Mostafa et al. [58]	2015	R	64	2	3.13	-	-
Lauritano et al. [59]	2016	R	87	1	1.15	0.23	0.019
Irani et al. [60]	2016	R	112	1	0.89	3.57	0.298
Varghese et al. [61]	2016	R	122	0	0.00	0.00	0.000
Yahalom et al. [76]	2016	P	30	3	10.00	3.70	0.308
Bandyopadhyay et al. [62]	2017	R	143	2	1.40	0.40	0.033
Gonzalez-Moles et al. [63]	2017	R	102	4	3.92	4.84	0.403
Rimkevičius et al. [64]	2017	R	133	3	2.26	-	-
Park et al. [65]	2018	R	113	0	0.00	0.00	0.000
Rock et al. [5]	2018	P	73	6	8.22	2.58	0.215
Laniosz et al. [66]	2019	R	303	7	2.31	-	-
Shearston et al. [19]	2019	R	281	4	1.42	0.31	0.026
Guan et al. [72]	2020	R	829	23	2.77	0.65	0.054
Arduino et al. [67]	2021	R	3173	82	2.58	-	-
Radochová et al. [68]	2021	R	271	2	0.74	0.14	0.011
Tsushima et al. [69]	2021	R	565	4	0.71	0.16	0.013
Zotti et al. [70]	2021	R	100	8	8.00	3.04	0.253
Cai et al. [71]	2022	R	3568	10	0.28	0.06	0.005
Total			24,277	331	1.36	0.32	0.027

R: retrospective; P: prospective; OSCC: Oral squamous cell carcinoma; MT: malignant transformation.

## Data Availability

The authors confirm that the data supporting the findings of this study are available from the corresponding author, upon reasonable request.

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
