# Peer review of "Rate of Malignant Transformation Differs Based on Diagnostic Criteria for Oral Lichenoid Conditions: A Systematic Review and Meta-Analysis of 24,277 Patients"

_cancers, 2023, doi:10.3390/cancers15092537_

Round 1
Reviewer 1 Report
With pleasure, I read the paper titled: "Rate of malignant transformation differs based on diagnostic 2 criteria for oral lichenoid conditions: A Systematic Review and 3 Meta-analysis of 24277 Patients". The topic is intellectually relevant to the journal Cancers. Collectively, the manuscript reads well and has proper flow of ideas and data are summarized adequately in pertinent tables and figures. The main strength of the paper includes being the most comprehensive report on the likelihood of malignant transformation for oral lichenoid. The limitations of the study and future directions are properly acknowledged. One of the best meta-analyses I have read recently. I just have a few minor questions:
1. Introduction. Please indicate if a previous meta-analysis has been done on the same topic. If no, please clearly highlight this significance. If yes, please highlight how your research is different and fills the literature gap.
2. Methods. For Newcastle-Ottawa scale, the maximum number of possible marks is 9 and not 8; please correct.
3. Results. The secondary meta-analyses based on subgroups are a big bonus. Have you done leave-one-sensitivity analyses to examine for robustness of effect sizes. It is not a mandatory change.
4. Discussion. Are there previous systematic review and meta-analysis projects on the same topic? If, please compare and contrast your findings.
Author Response
Please see the attachment. Many thanks.

Reviewer 2 Report
Dear authors,
thank you for bringing once again this important topic to our attention. The data collection is well performed, the presentation/description of the graphs might be graphically improved, and some of the tables moved to the supplement.
The biggest shortfall to me seems the discussion:
regarding your primary objective (i.d.: the incidence btw. the 2 different definitions): you are using the wording from the Gonzales-Moles (61) group`s abstract, but do not go into the real interesting depth about the two different classifications and what was said/found by other authors. I encourage you to add a few more sentences on this topic. Also, you do not give any good reason, why hypertension was included/any bilogical rational. And I miss the discussion of gender dimorphism/comparison with other authors etc.
I appreciate the suggestions for future use of AI based approaches for monitoring, as this may really be a game changer for these patients.
Concerning language: there are minor /speling/syntax/table labelling mistakes. Please, try to correct them.
Please, find an annotated PDF file attached.
Author Response
Please see the attachment. Many thanks.
